# OpenReview forum: "Library Hallucinations in LLMs: Risk Analysis Grounded in Developer Queries"
_ICLR.cc/2026/Conference — Submitted to ICLR 2026_

### Official Review · Reviewer_BPXC · 2025-11-01

**Soundness:** 3
**Presentation:** 4
**Contribution:** 3
**Rating:** 6
**Confidence:** 4

**Summary:**

This paper presents an empirical analysis of the extent to which language models for code hallucinate concerning dependent software libraries.

The paper investigates six language models (GPT4, GPT5, Ministral, Qwen, Llama, and DeepSeek). The dataset used starts from BigCodeBench, yielding a dataset of 356 tasks using 30 distinct libraries. Three experiments are conducted:

1. To what extent do variations in user descriptions of libraries to be used influence hallucinations?
2. To what extent do misspellings in prompts lead to hallucinations?
3. To what extent do prompt engineering techniques (such as chain of thought, or step back) lead to hallucinations?

The experiments find that:

1. Adjectives (like fast, modern) to libary instructions generally have no influence; adding a year ("from 2025") does increase presence of hallucinations
2. Simple spelling mistakes leads to hallucinations in up to 26% of the tasks, with all LLMs resorting to fake library names.
3. Prompt engineering techniques like self-analysis and explicit checks can reduce hallucinations, but chanin of thought and step-back prompting often worsen hallucination rates.

The paper discusses how asking for lesser known libraries can further increase hallucinations, and whether hallucinations could be considered as sources of inspiration for desired functionality.

**Strengths:**

- A very well written paper
- Rigorous, well conducted experiment
- Clear and understandable results from the experiments
- A relevant problem

**Weaknesses:**

The findings are clear and the experiments are very well conducted. I was, however, somewhat disappointed by the discussion, which seems to accept the results at face value and just presents two somewhat unrelated ideas. I was hoping for a more rigorous treatment of underlying principles, consequences and implications, also to strengthen the sense of urgency from this paper. Why are we getting these results? What should be done about them?

For example, the finding that typos can have detrimental effects is a nice one -- but how could it be addressed? Could it be circumvented by having a tool check anything that might be a library, and spellcheck the prompt to use correct library names? Or: could LLMs be made more robust by adding 'adversarial' finetuning with likely typos?

The setup could also be viewed as a way to evaluate the effect of countermeasures. In practice, new library versions will continue to arrive after an LLM has been created, so RAG-based solutions or frequent fine tuning approaches are inevitable. But how well would they work? Might the setup proposed in this paper help?

**Questions:**

- What would you include in a bolder, more urgent discussion section?

---

> ### Author Response · Authors · 2025-11-21
>
> We would like to thank you for your positive feedback. We are greatly encouraged by your comments about the quality of our writing, the rigor of our experiments, the clarity of our results and the relevance of the problem.
>
> *To address your concerns we have made the following revisions: added practical implications and actions section (p8), added initial fine-tuning experiment (Appendix E.1), emphasised our benchmark dataset (Section 7).*
>
> Below, we hope to comprehensively address your main question, along with the other concerns raised in your comments.
>
> ***In response to your main question “What would you include in a bolder, more urgent discussion section?” and first comment:***
>
> Great question! We would discuss how our findings impact developers in their day to day workflows, the threats they are exposed to, that we need more than simple prompt-engineering strategies to mitigate this serious problem, and importantly that these mitigations need to be implemented at the model interface and not at the user level. We have now added “Section 5.1 Practical Implications & Actions” to cover this aspect of the paper, drawing more attention to, and raising the urgency of, the issues we raise.
>
> ***In response to your additional comments:***
>
> Thank you for your insightful comments about potential mitigations. As we discuss throughout the paper, our focus is on realistic developer scenarios, and assisting LLM users in their ability to avoid library hallucinations. While external tools can help to mitigate the problem, in practice it is likely that developers will just continue to use the latest base models via their chosen interface. Therefore, we believe LLM creators need to implement safeguards at the model building stage, before release, to ensure they are in place for all developers.
>
> Regarding finetuning, we have now added some preliminary results on using fine-tuning as a mitigation method for library name hallucinations triggered by user mistakes. As you suggested, we use training data that encourages the model to correct any typos/mistakes in the library name. Full details are given in Appendix E.1.
>
> We appreciate your comments regarding the setup in the paper being used for future evaluation of new tools and mitigations. We realise now that we did not emphasize it enough in the main text, but we have released a benchmark dataset and evaluation pipeline (using the set up from our experiments) that can be utilised for exactly this purpose. We have added “Section 7 LibraryHalluBench” to fully describe this contribution of our paper, and ensure our processes can be used for future investigations into library-related hallucinations.

---

> ### Author Response · Authors · 2025-11-25
>
> Dear Reviewer BPXC,
>
> We hope our responses and revisions have addressed your concerns. If so, we would greatly appreciate it if you could consider updating your scores accordingly.
>
> Otherwise, if there are any remaining issues we haven't fully addressed, we're happy to provide further clarification and revisions before the discussion period closes.
>
> Thank you again for your time and thoughtful feedback!

---

### Official Review · Reviewer_KMKa · 2025-11-04

**Soundness:** 2
**Presentation:** 3
**Contribution:** 2
**Rating:** 2
**Confidence:** 4

**Summary:**

This paper presents the first systematic study of how user-level prompt variations impact library hallucinations in LLM-generated code, evaluating six diverse LLMs (GPT-4o-mini, GPT-5-mini, Ministral-8B, Qwen2.5-Coder, Llama-3.3, DeepSeek-V3.1) across 356 Python coding tasks from BigCodeBench. The authors investigate two hallucination types, library name hallucinations and library member hallucinations, through three systematic experiments examining realistic user language from developer forums, user errors of varying severity, and prompt engineering mitigation strategies. Key findings reveal systemic vulnerabilities: while adjective-based descriptions like "fast" or "lightweight" are largely ignored, year-based prompts asking for libraries "from 2025" trigger hallucinations in up to 84% of tasks; one-character misspellings cause hallucinations in up to 26% of tasks, multi-character misspellings in up to 79%, and fake library names are accepted in up to 99% of tasks, demonstrating extreme model sycophancy. Prompt engineering shows mixed results, self-analysis and explicit-check strategies reduce hallucinations in ~80% of cases, while popular reasoning prompts like cot often increase hallucination rates.

**Strengths:**

1. The authors provide a precise definition of "library hallucinations" as two types of measurable failures (non-existent package names and invalid member calls from valid packages), and systematically measure them using three groups of realistic prompt factors (descriptive words/year-based prompts, user spelling errors, low-barrier prompt engineering). The narrative structure is clear and well-organized.

2. The experimental design is rigorous and comprehensive: it evaluates 6 diverse LLMs (covering open/closed source, general/code-specific, 8B-671B parameters) on 356 carefully filtered tasks from BigCodeBench, generating 3 responses per task using fresh API sessions to avoid caching effects.

3. The paper establishes a particularly insightful novel connection between LLM hallucinations and supply chain attacks (typosquatting/slopsquatting), revealing how models' sycophantic behavior, accepting fake library names in up to 99% of tasks, creates exploitable vulnerabilities.

4. The paper systematically compares four low-cost mitigation strategies (CoT, Step-back, Self-analysis, Explicit-check), providing immediately usable "lightweight prompt modification" baselines for practical engineering teams.

**Weaknesses:**

1. The Python-only scope with merely 30 libraries may not represent broader ecosystems. Results may not generalize to JavaScript, Java, or Rust with different packaging conventions, and the paper does not analyze whether findings vary by library popularity, domain, or API complexity, despite BigCodeBench containing 7 distinct domains.

2. The hallucination detection methodology is problematic: checking only against the latest documentation fails to distinguish true hallucinations from outdated knowledge or version mismatches, which is critical since models trained on older versions may correctly reference deprecated APIs. The manual review process lacks details on inter-rater reliability.

3. The most fundamental problem is the lack of methodological innovation, this is primarily a measurement study that does not propose any novel solutions. The paper only tests 4 existing prompt engineering techniques (chain-of-thought, step-back, etc.), which essentially reflects an absence of methodological contribution. For instance, the interesting finding in Experiment 3 that chain-of-thought prompting increases hallucination rates in 50% of cases is not leveraged to design better methods.

**Questions:**

See the section above.

---

> ### Author Response · Authors · 2025-11-21
> **Initial response part 1**
>
> Thank you for your detailed review, we sincerely appreciate the valuable and constructive feedback. We are encouraged by your positive comments about the clarity, rigor and comprehensiveness of our study.
>
> *To address your concerns we have made the following revisions: added generalisability experiment (Appendix E.2), added domain analysis of results (Appendix D), emphasised our benchmark dataset (Section 7), added initial fine-tuning experiment (Appendix E.1), added more detail regarding library version handling (L297-302) and manual analysis (L266-268).*
>
> Below we provide further details and comprehensive answers to your comments. We look forward to hearing back from you if you have any further comments or questions.
>
> ***Regarding “The Python-only scope with merely 30 libraries may not represent broader ecosystems.”***
>
> Thank you for your valuable feedback on the reliability of our results. In the study, we deliberately focused on a single language and benchmark to allow us to go in depth with our analysis, and investigate many prompt variations, instead of doing a shallow cross-language sweep. We believe this approach has led to some interesting findings, but agree that a key limitation and important piece of future work is ensuring the generalisability of the results.
>
> Due to time constraints in the discussion period we are unable to add additional results for other languages (requiring an additional data extraction and evaluation pipeline), instead we focussed on adding an initial investigation into the generalisability of our results over the wider Python ecosystem. We repeat the experiments that trigger the highest hallucination rates over a subset of our models, for the CodeInsight dataset. Similar hallucination rates are seen, indicating that our findings from the main paper generalise across a wider range of Python tasks and libraries. Full experiment details are available in Appendix E.2.
>
> ***Regarding “...the paper does not analyze whether findings vary by library popularity, domain, or API complexity…”***
>
> Thank you for this insightful suggestion, we agree that more analysis could be useful and allow for even more insight into the results. We have now conducted a domain analysis, where we take the results from experiments 1 and 2 and look at the hallucination rates over the 7 different domains included in BigCodeBench. The results are contained in Appendix D. We observe a trend towards higher hallucination rates in the more niche domains.
>
> ***Regarding “checking only against the latest documentation fails to distinguish true hallucinations from outdated knowledge or version mismatches”***
>
> Thank you for the constructive feedback, we apologise for not including more detail on how we handled library versions in the original writing of the paper. We used the documentation from the latest version because that is the default version installed when a version is not specified; we believe it is valid for an LLM to suggest members from previous library versions, but only if they specify the version.
>
> In our hallucination detection pipeline, we programmatically checked each response for the version string of any valid version of the corresponding library, and only registered member hallucinations if no version was included. This was conservative, but LLMs explicitly specify a library version incredibly rarely, so this made almost no difference to the results. We have included more information on this in our revision (L297-302).
>
> ***Regarding “The manual review process lacks details on inter-rater reliability.”***
>
> Thank you for your comment, we apologise that this process was not better described in the paper. Any manual analysis was performed by a single author, by performing web searches looking for import examples, therefore there is no inter-rater reliability. We have added this additional information into the paper (L266-268).

---

> > ### Author Response · Authors · 2025-11-21
> > **Initial response part 2**
> >
> > ***Regarding “The most fundamental problem is the lack of methodological innovation, this is primarily a measurement study that does not propose any novel solutions.”***
> >
> > Thank you for your comment, we appreciate your feedback on the style of our study. Our primary contribution is methodological on the measurement side rather than proposing a new algorithmic solution, and we do believe there is novelty: we have operationalised the definition and measurement of “library hallucinations”; we design realistic prompts via the systematic analysis of real user queries; we consider a range of severity in the mistakes that we use to prompt the LLMs.
> >
> > Additionally, we provide a novel benchmark and evaluation pipeline that allows our experiments to be used to measure hallucinations in other models. We have further emphasized this contribution in the main paper (Section 7 LibraryHalluBench), as we now realise its availability was unclear.
> >
> > We deliberately left the design of novel solutions as future work, and focussed our study on measurement of library hallucinations to allow a thorough and in-depth analysis of what we believe is an important problem for real-world developers. This is what inspired our investigation into only prompt-engineering mitigations; we wanted to check techniques that are low-cost, practical and already available to users.
> >
> > To encourage future work into using more sophisticated techniques to reduce library hallucinations, we have conducted an initial investigation into fine-tuning as a mitigation method for library name hallucinations triggered by user mistakes. We use training data that encourages the model to correct any typos/mistakes in the library name, full details and results are given in Appendix E.1.

---

> ### Author Response · Authors · 2025-11-25
>
> Dear Reviewer KMKa,
>
> We hope our responses and revisions have addressed your concerns. If so, we would greatly appreciate it if you could consider updating your scores accordingly.
>
> Otherwise, if there are any remaining issues we haven't fully addressed, we're happy to provide further clarification and revisions before the discussion period closes.
>
> Thank you again for your time and thoughtful feedback!

---

### Official Review · Reviewer_DLB3 · 2025-11-06

**Soundness:** 3
**Presentation:** 2
**Contribution:** 3
**Rating:** 4
**Confidence:** 2

**Summary:**

This paper conducts the comprehensive analysis of how realistic prompt variations affect hallucinations of library names and functions in LLM-generated Python code. The authors discover that while descriptive adjectives in prompts have little effect, subtle phrasing changes—such as year-based descriptions, or one-character misspelling-can significantly increase hallucination frequency. Although some prompt-engineering strategies can reduce these errors, their effectiveness is not consistent. The future work should focus more on this issue for building a more reliable coding model.

**Strengths:**

(s1) The studied problem is important and realistic.

(s2) The findings in this paper may provide some new insights to the community.

(s3) The empirical results support some claims in the paper.

**Weaknesses:**

(w1) The writing can be improved. For example, a figure of general description can be provided for the better understanding of readers.

(w2) I would like to see the evaluation results of more advanced coding LLMs, such as Claude. The selected models in the current version do not yet represent the best available coding capability..

(w3) The authors only conduct evaluations on Python language on one specific dataset (BigCodeBench). More results on other datasets would strengthen the paper.

(w4) The prompting strategies show limited effectiveness in mitigating the hallucinations. I would suggest the authors try some training-based method.

(w5) The Broader Impact section is missing, and I think this paper should include a discussion on some potential social impacts.

**Questions:**

See the weaknesses.

**Details Of Ethics Concerns:**

The findings of this paper might potentially be exploited for malicious purposes, posing risks in LLM-based code generation scenarios.

---

> ### Author Response · Authors · 2025-11-21
>
> We thank Reviewer DLB3 for their detailed review and constructive feedback, we appreciate that you find our studied problem important and realistic, we also believe it can provide insights to the community.
>
> *To address your concerns we have made the following revisions: added a figure (p2), added full results for Claude-4.5-Haiku (p6,7,9), added generalisability experiment (Appendix E.2), added initial fine-tuning experiment (Appendix E.1), added practical implications and actions section (p8).*
>
> Below we provide further details and comprehensive answers to your comments.
>
> ***Regarding “(w1) The writing can be improved. For example, a figure…”***
>
> Thanks for your insightful feedback, we agree that a simple figure to help general understanding of our process is a welcome addition. A figure has now been added to page 2 of the revised paper. We have also made an effort to improve the writing throughout where possible, but if you have any further comments on specific improvements that could be made to the writing then we would be more than happy to resolve them before the end of the discussion period.
>
>
> ***Regarding “(w2) I would like to see the evaluation results of more advanced coding LLMs, such as Claude.”***
>
> We really appreciate your insight here, it certainly felt like there was an almost unlimited number of LLMs that we could have chosen when conducting the study, and we tried to choose a broad range whilst minimizing cost. That being said, we agree that adding a Claude model is a great idea, and we have now added full results for Claude 4.5 Haiku to the paper. The results show that Claude is one of the LLMs most sensitive to adjective-based descriptions, but had the least hallucinations from year-based descriptions; for user mistakes it showed high hallucination rates similar to the other state-of-the-art LLMs (GPT5 and DeepSeek). In general, adding Claude brings a nice balance to the model selection, allowing comparison to other frontier closed-source models from OpenAI.
>
>
> ***Regarding “(w3) The authors only conduct evaluations on Python language on one specific dataset…”***
>
> Thank you for your constructive feedback on our evaluations. In the study, we deliberately focused on a single language and benchmark to allow us to go in depth with our analysis, and investigate many prompt variations, instead of doing a shallow cross-language sweep. We believe this approach has led to some interesting findings, but agree that a key limitation and important piece of future work is ensuring the generalisability of the results.
>
> Due to time constraints in the discussion period we are unable to add additional results for other languages (requiring an additional data extraction and evaluation pipeline), instead we focussed on adding an initial investigation into the generalisability of our results over the wider Python ecosystem. We repeat the experiments that trigger the highest hallucination rates over a subset of our models, for the CodeInsight dataset. Similar hallucination rates are seen, indicating that our findings from the main paper generalise across a wider range of Python tasks and libraries. Full experiment details are available in Appendix E.2.
>
>
> ***Regarding “(w4) … I would suggest the authors try some training-based method.”***
>
> We are thankful for your insight here, and agree with your sentiment that more sophisticated strategies are needed after the prompt strategies appear to provide minimal assistance. We wanted to keep our main study grounded in techniques that real developers could employ in their day to day workflow, which is why we chose to focus on prompt-engineering and didn’t include any training approaches initially. But we agree that it is an important avenue of future work.
>
> We have now added some preliminary results on using fine-tuning as a mitigation method for library name hallucinations triggered by user mistakes. We use training data that encourages the model to correct any typos/mistakes in the library name. Full details are given in Appendix E.1, we hope this preliminary investigation will help to inspire future work in this area.
>
>
> ***Regarding “(w5) The Broader Impact section is missing…”***
>
> We appreciate your constructive suggestion for a broader impact section. We tried to cover the practical implications throughout the paper (in both the Introduction and the Results), but we agree that an explicit section of the paper dedicated to this topic is important. We have now added “Section 5.1 Practical Implications & Actions” to address this.

---

> ### Author Response · Authors · 2025-11-25
>
> Dear Reviewer DLB3,
>
> We hope our responses and revisions have addressed your concerns. If so, we would greatly appreciate it if you could consider updating your scores accordingly.
>
> Otherwise, if there are any remaining issues we haven't fully addressed, we're happy to provide further clarification and revisions before the discussion period closes.
>
> Thank you again for your time and thoughtful feedback!

---

### Author Response · Authors · 2025-11-21
**Dear Reviewers DLB3, KMKa and BPXC, and Area Chair,**

We sincerely appreciate the time and effort you have dedicated to reviewing our work. We thank all reviewers for their thoughtful and constructive feedback, and are encouraged by the many positive comments also given. We have responded to all of your queries in your individual responses below, and have revised the paper to address all points raised.

A summary of all enhancements made to the paper during the discussion period, inspired by your valuable comments, is as follows:

- Added a diagram to aid in general understanding of our pipeline. (p2)
- Included full results for Claude-4.5-Haiku. (p6,7,9)
- Emphasized the published benchmark dataset, that allows readers to evaluate their own models, in the main text. (p10)
- Added a discussion of the practical implications and broader impacts. (p8)
- Domain analysis appendix, showing the results from the main paper aggregated over the task domains. (p25)
- Finetuning appendix, containing an initial investigation into how a simple finetuning strategy could help mitigate hallucinations. (p25)
- Generalisability appendix, initial results indicating that our findings generalise to other real-world tasks, and a wider range of libraries. (p26)
- Various improvements to the writing, with added detail for certain topics, such manual result analysis (p5) and version checking (p6).

We look forward to hearing more from you, and further discussing our paper and the revisions we have made.

---

### Author Response · Authors · 2025-11-30
**Dear New Area Chair,**

Thank you for stepping in after the difficult recent events, we sincerely appreciate the extra time and effort required from you to process all of the reviewers comments and our responses.

We’ve worked hard during the discussion phase to substantially strengthen the paper and ensure all revisions are clear, traceable, and easy to verify. We have addressed each and every one of the reviewers’ comments, as summarised below.

**Regarding DLB3 point w1** - *add a figure to improve clarity*
- We have added a simple figure (page 2), to help readers to quickly gain a general understanding of our process.

**Regarding DLB3 point w2** - *add Claude results*
- We have added full results for Claude-4.5-Haiku, and included discussion of them throughout. The results can be seen in Tables 1, 2 and 3 (pages 6, 7, and 9 respectively).
- We observe similar hallucination patterns for Claude: it is one of the most sensitive to adjective-based descriptions; one of the least sensitive to year based prompts; and had high hallucination rates when faced with user mistakes.

**Regarding DLB3 point w3, and KMKa point 1** - *results only cover a single dataset*
- We have added results for another dataset in Appendix E.2 (pages 26 and 27). We have used the CodeInsight dataset (seeded from StackExchange) and an expanded range of libraries for this, the results indicate that our findings generalise to the wider Python ecosystem.

**Regarding DLB3 point w4, and BPXC comment** - *could a training-based method help*
- We have added results for an experiment investigating how finetuning could mitigate the observed hallucinations, in Appendix E.1 (pages 25 and 26).

**Regarding DLB3 point w5, and BPXC question** - *there could be a broader discussion of the implications*
- We have added a focused ‘Practical Implications & Actions’ section (page 8) that connects findings to real developer workflows, concrete risks, and the urgent need for model-side safeguards.

**Regarding KMKa point 1** - *lacking domain analysis*
- We have added a full analysis of the results aggregated over the different task domains contained within BigCodeBench, in Appendix D (pages 25 and 26).

**Regarding KMKa point 2** - *hallucination detection comments*
- We have expanded our discussion of how we handle deprecated library features (both when conducting experiments L297-302, and when analysing the results L345-349) - we automatically check for mentions of any corresponding library version in the response, and take a conservative approach of never counting a hallucination if a library version is present.

**Regarding KMKa point 2** - *manual review process*
- We have better described our manual analysis of hallucinations (L266-268) - performed by a single author, via a web-based search.

**Regarding KMKa point 3** - *"The most fundamental problem is the lack of methodological innovation, this is primarily a measurement study that does not propose any novel solutions."*
- We agree that our paper is primarily a measurement study, but we believe this is exactly where its value lies. As stated in the ICLR reviewer guide, submissions can contribute meaningfully by “convincingly demonstrating new, relevant, impactful knowledge,” and this explicitly includes empirical work, so long as it is “scientifically rigorous” in supporting the claims made. Our goal is to provide precisely this kind of contribution.
- Our contribution lies in (1) designing the first systematic framework for understanding library hallucinations across two hallucination types with realistic prompts grounded in developer queries, and (2) uncovering previously unknown behavioral vulnerabilities that persist across different LLMs (such as typosquatting, as recognised by Reviewer KMKa themself). Our findings have immediate practical relevance for developers (e.g., how to structure prompts to reduce hallucination risks) and for LLM builders (e.g., actionable directions for improving training, decoding, and safety mechanisms to mitigate library hallucinations).
- ***We view this as foundational science: before new solutions can be proposed, the field needs a precise understanding of the failure landscape.***
- Additionally, we provide a novel benchmark and evaluation pipeline that allows our experiments to be used to measure library-related hallucinations in other models. We have further emphasized this contribution in the main paper by adding Section 7 LibraryHalluBench (page 10), as we now realise its availability was unclear.

**Regarding BPXC comment** - *can the setup help with future evaluation*
- We have emphasized the benchmark dataset and evaluation pipeline that we provide publicly in our GitHub repository, for this very purpose. We did this by adding Section 7 LibraryHalluBench (page 10) to fully explain our contributions.

Thank you again for your hard work throughout this difficult process. Please let us know if any further adjustments would be helpful before the discussion period closes.

---

### Meta-Review · Area_Chair_NhCS · 2026-01-02

**Summary:**

This paper studied about how real-world prompt variations affect hallucination rate. It defines two type of hallucinations libraray name hallucinations and library member hallucinations. Through evaluations,  It reveals some vulnerabilities such one-character misspellings in library names trigger hallucinations in up to 26% of tasks, fake library names are accepted in up to 99% of tasks, and time-related prompts lead to hallucinations in up to 84% of tasks.

Overall, AC feels most the reviewers share the similar concern such as python-only evaluation, lack discussion on mitigation strategies, innovation, and deeper treatment of implications.
AC feels it is ok the sell the contribution as benchmark paper and likes the analysis of this paper. AC thinks most of the reviewers' concerns are addressed. But some of the important concerns from reviewers such as  more language and deeper treatment of implications are not fully addressed (more details in the reviewer concerns section). These aspects are important, the main contribution of this paper is from benchmark perspective.

**Reviewer Concerns:**

Reviewer DLB3's conerns:
1). Lack of a high-level figure and clarity of presentation.
Author addressed it well
2). Missing evaluation on stronger coding models (e.g., Claude)
Author added claude model.
3). Results limited to a single dataset and python.
Author added more datasets. AC understands the scope of this paper is restricted to Python but adds more evaluation on other language will significantly increase the impact of this paper.
4). Try some training-based methods.

The authors added training-based experiments. However, there is no brief/clear summary of whether the approach works (under what conditions) in the main context of the paper Also, there is no discussion on what these results imply for designing stronger future defenses/mitigations.



Reviewer KMKa's concerns

1) Python-only scope and a limited number of libraries
Similar to the previous reviewer, the concern is partially addressed.

2)The hallucination detection methodology is problematic:
Authors have added new results.
3) manual review process.
Authors have added details.

4). methodological innovation.
Authors have explained the innovation novelty from the benchmarking perspectives.



Reviewer BPXC's Concerns
(1) Discussion lacked urgency and deeper treatment of implications.
Authors added some discussions in Sec 5.1 and conducted experiment about the fine-tuning based methods.

Overall, AC feels most reviewers share a similar concern, such as python-only evaluation, lackof  discussion on mitigation strategies, innovation, and deeper treatment of implications.
AC feels it is ok the sell the contribution as a benchmark paper and like the analysis of this paper and thinks most of the reviewers' concerns are addressed.  But some of the important concerns from reviewers such as more language and deeper treatment of implications are not fully addressed (more details in the reviewer concerns section). These aspects are important as the main contribution of this paper is from a benchmark perspective.

**Reviewer Scores:**

Reviewer DLB3 may not be positive about this paper since the concerns such as python-only, implication are not fully addressed.
Reviewer KmKa should increase the score to 4 at least (may not be positive) since the python-only concern is not fully addressed.
Reviewer BPXC will keep the score.

---

### Decision · Program_Chairs · 2026-01-26

Reject